# Zeolitic Imidazolium Frameworks-Derived Ru-Based Composite Materials Enable the Catalytic Dehydrogenation of Alcohols to Carboxylic Acids

Zhan Chen, Jing Hang, Song Zhang, Ye Yuan *, Francis Verpoort * and Cheng Chen *

State Key Laboratory of Advanced Technology for Materials Synthesis and Processing, Wuhan University of Technology, 122 Luoshi Road, Wuhan 430070, China; 317376@whut.edu.cn (Z.C.); 303735@whut.edu.cn (J.H.); zhangsongzzss@whut.edu.cn (S.Z.)

* Correspondence: fyyuanye@whut.edu.cn (Y.Y.); francis@whut.edu.cn (F.V.); chengchen@whut.edu.cn (C.C.)

**Abstract:** The metal-catalyzed dehydrogenation of alcohols without any oxidant or additive has been demonstrated as an atom-economic and environmentally friendly strategy for carboxylic acid synthesis. Among the various catalysts applied to this transformation, Ru-based homogeneous ones have been the most extensively studied owing to their remarkable catalytic activity. However, these catalysts required multiple complicated synthesis steps. In addition, they were either difficult to recycle or their recovery processes were relatively tedious. Therefore, a series of Ru-containing heterogeneous catalysts with zeolitic imidazolium frameworks (ZIFs)-derived materials were designed and fabricated. A thorough screening of various parameters was conducted, and it was found that the material obtained by loading a Ru concentration of 0.05 mol/L into Co species embedded in porous N-doped carbon ($Ru_{0.05}$@Co-NC) had the best catalytic performance in this transformation, affording a handful of carboxylic acid products from the corresponding aromatic or aliphatic alcohols in moderate to high yields. Additionally, the catalyst showed remarkable recyclability as it could be recycled eight times with stable activity fluctuation (45–52%). It is noteworthy that catalyst recycling was convenient and fast, which could be realized simply by an external magnet. Moreover, the stable morphology and structure of $Ru_{0.05}$@Co-NC, along with its high specific surface area, hierarchical pore structure, high porosity, and other properties, jointly contributed to its high catalytic activity and good recyclability. Furthermore, the stability and activity of $Ru_{0.05}$@Co-NC were further evaluated through acid etching experiments, which revealed that some Ru species could stably exist in concentrated acids and play a pivotal role in promoting this catalytic process.

**Keywords:** zeolitic imidazolium frameworks (ZIFs); metal species embedded in N-doped carbon (M-NC); ruthenium-based heterogenous catalyst; alcohol dehydrogenation; carboxylic acid synthesis

## 1. Introduction

Carboxylic acids undergo conversion into various chemicals, such as acyl halides, anhydrides, esters, and amides [1], and these derivatives possess distinctive characteristics and are widely utilized in the chemical industry [2]. Therefore, versatile synthetic approaches have been successively developed for their synthesis [3,4]. Among these methods, the metal-catalyzed conversion of primary alcohols into the corresponding carboxylic acids has been considered a general and efficient transformation. In general, this process occurs in the presence of external oxidants [4,5], which makes it undesirable in terms of greenness and sustainability. Even though some strategies employ dioxygen gas as a mild oxidant [6–8], operational safety and functional group tolerance continue to limit the process. To avoid the possible adverse effects of additional oxidants in the reaction process, the acceptorless dehydrogenative coupling of alcohols and water/hydroxides, with dihydrogen gas as the only byproduct, has been regarded as an atom-economic and environmentally friendly protocol for carboxylic acid preparation. Inspired by the catalytic transfer

hydrogenation of alcohols and water with the assistance of hydrogen acceptors [9–11], the Milstein group developed an acceptorless way of this catalysis for the first time [12] by capitalizing on the remarkable performance of catalytic dehydrogenation procedures. In this work, a handful of primary alcohols were swiftly converted to carboxylic acids under hydrogen acceptor-free circumstances, utilizing their well-known pincer-type Ru complex in aqueous NaOH solutions. Subsequently, a variety of metal-based catalytic systems were successively developed [13–33]. Among the above-mentioned systems, the Ru-based ones are the most extensively explored owing to the remarkable catalytic potency of Ru in this process. It was noted that most of these Ru catalysts were homogenous [34–41], but a relatively complicated synthesis (due to the employment of privileged ligands and the subsequent coordination with Ru sources) was adopted to achieve satisfactory results. Even though several homogeneous Ru catalysts could be recycled for a few cycles (≤5), centrifugation/suction filtration was necessary to recover the catalysts, while the re-activation of the recovered catalysts was also required for the next cycle. Therefore, the construction of robust Ru catalysts featuring both simple synthetic routes and facile recycling techniques is still in high demand.

According to published studies, the Lewis acid site-basic site pair is essential for catalyzing the dehydrogenation of primary alcohols to carboxylic acids in an efficient manner [22,23]. In general, metal sites are responsible for providing Lewis acid sites; thus, Ru would be chosen as the ideal candidate due to its extraordinary catalytic performance in this transformation. The key issue lies in the selection of appropriate materials possessing the required basic sites. Recently, zeolitic imidazolium frameworks (ZIFs) have received considerable attention due to their characteristics, including their zeolite-like structure, large specific surface area, and ordered porosity [42]. Most significantly, this class of materials possesses the abundant basic sites required for our intended transformation. However, the stability issues of ZIFs limit their further applications under harsh conditions, such as high temperatures and strong acidic/alkaline media [43]. It has been well documented that controlled thermal annealing is an effective technique for addressing the stability issues of ZIFs. By using this method, it is possible to convert ZIFs into metal/metal oxides embedded in porous N-doped carbon (M-NC), which not only retains the favorable characteristics (the basic sites) of ZIFs, but also achieves higher stability than their parent materials [44,45]. In particular, Co-NC materials derived from ZIF-67 have captured our great attention due to the following two aspects [18,23]: (1) they provide abundant amounts of basic sites and (2) their magnetic properties can facilitate the procedure of catalytic recycling. Thus, Co-NC materials were chosen to supply the basic sites for our designed catalysts.

After selecting the Lewis acid site-basic site pair for this study, we intended to design and fabricate a series of Ru@Co-NC materials via a rational combination of a Ru precursor and ZIF-67-derived Co-NC. Moreover, the impact of catalyst structure, Ru loadings, active sites, reaction parameters, etc., on the catalytic dehydrogenation of alcohols to carboxylic acids was thoroughly explored. After extensive optimization, the best-performing catalyst ($Ru_{0.05}$@Co-NC) was identified as a highly active catalyst for this transformation, efficiently converting a handful of primary alcohols into the corresponding carboxylic acids. It was worth noting that the catalyst exhibited remarkable recyclability since around 50% of yields could be retained for eight consecutive cycles without an obvious activity decline. Except for the facile synthetic route, $Ru_{0.05}$@Co-NC also featured a simple recycling technique, as only an external magnet was utilized to recover the catalyst and reuse it in the next cycle, without additional treatment or activation. Furthermore, the stability and activity of $Ru_{0.05}$@Co-NC were evaluated through a series of characterization techniques and acid etching experiments. A comparison of the present catalyst with the reported heterogeneous catalysts in the literature was outlined in Table S1. Compared with the reported noble metal-based catalysts, $Ru_{0.05}$@Co-NC features the lowest metal loading and most recycle bounds. It is also worth noting that the product yield could retain around 50% for eight consecutive recycle bounds. In comparison with the non-noble metal catalysts in the

literature, $Ru_{0.05}$@Co-NC could promote this transformation either under milder reaction conditions or with a lower catalyst amount.

## 2. Results and Discussion

### 2.1. Synthesis and Characterization of $Ru_x$@Co-NC

To achieve high-performance Ru-based catalysts for this dehydrogenation process, it is crucial to incorporate a Lewis acid site-basic site pair into their design. Accordingly, we intended to construct a series of $Ru_x$@Co-NC materials by rationally combining a Ru precursor as the Lewis acid sites and ZIF-67-derived Co-NC as the basic sites. The synthetic procedure of $Ru_x$@Co-NC was illustrated in Figure 1. Firstly, the ZIF-67 precursor was designed and synthesized by mixing $Co(NO_3)_2 \cdot 6H_2O$, 2-MIM, and methanol at room temperature, so that the metal ions ($Co^{2+}$) could coordinate with the imidazole linker (2-MIM) to form a well-ordered framework. Subsequently, the precursor material was pyrolyzed under controlled conditions to afford the corresponding Co-NC. Finally, the designed $Ru_x$@Co-NC materials were obtained from the treatment of Co-NC with a Ru precursor ($RuCl_3 \cdot 3H_2O$) via a chemical reduction reaction, in which $Ru^{3+}$ was reduced to $Ru^0$ by $Co^0$ in Co-NC species. After the synthesis of $Ru_x$@Co-NC, a series of characterization techniques were used to confirm the successful incorporation of Ru into Co-NC, with $Ru_{0.05}$@Co-NC as a representative example for the investigations. Moreover, the structural, morphological, and porous properties of $Ru_{0.05}$@Co-NC and Co-NC were analyzed and compared to have a complete understanding of their properties.

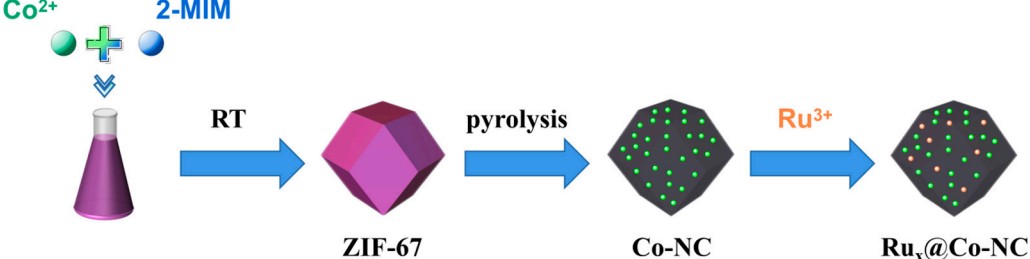

**Figure 1.** The process of $Ru_x$@Co-NC preparation.

The X-ray diffraction (XRD) diffractograms of Co-NC and $Ru_{0.05}$@Co-NC indicated that the two materials displayed similar patterns, revealing that the desired structure in Co-NC was successfully retained in $Ru_{0.05}$@Co-NC (Figure 2a). Among the peaks at three different positions in the diffractogram of Co-NC, 44.2°, 51.5°, and 75.8° corresponded to the crystalline planes Co (111), Co (200), and Co (220), respectively, which is consistent with the data in a previous study [46]. Compared with Co-NC, the positions in the diffractogram of $Ru_{0.05}$@Co-NC (44.1°, 51.1°, 75.7°) were slightly shifted. Moreover, the lack of detectable Ru peaks may be due to a low Ru concentration, the amorphous nature of the material, or the sub nanoscopic Ru particle size. To further confirm the existence of Ru in our desired material, additional investigations were performed. The chemical valences of different elements in $Ru_{0.05}$@Co-NC were examined by X-ray photoelectron spectroscopy (XPS, Figure 2b), and specifically, C–O at 287.5 eV, C=N at 285.5 eV, and C–C at 284.7 eV were detected in the C 1s spectra. More importantly, the characteristic peak of $Ru^0$ 3d (282.1 eV) was found to be nearby the above-mentioned C 1s signals, indicating that $Ru^0$ was successfully integrated into the material (the surface composition was shown in Table S2). Additionally, the inductively coupled plasma emission spectroscopy (ICP-OES) results indicated that 3.8 wt% of the Ru element was found in $Ru_{0.05}$@Co-NC (Table S3). The presence of acid and basic sites has been reported to be crucial for achieving excellent catalytic performance of catalysts [23]. Hence, the acid and basic characters of $Ru_{0.05}$@Co-NC were investigated by temperature programmed desorption (TPD) analysis, using $NH_3$ and $CO_2$ as the probe gas, respectively (Figure 2c,d). From the $CO_2$-TPD result, a strong peak at 467 °C corresponding to $CO_2$ desorption from the respective strong basic sites was

mainly detected (Figure 2c). It is also worth noting that the total amounts of basic sites were determined as 6.94 µmol/g. On the other hand, the $NH_3$-TPD profile suggested that a weak peak at 201 °C and a strong signal at 475 °C were observed, indicating weak and strong acid sites, respectively (Figure 2d). In addition, the total acid sites were quantitatively described to be 9.12 µmol/g.

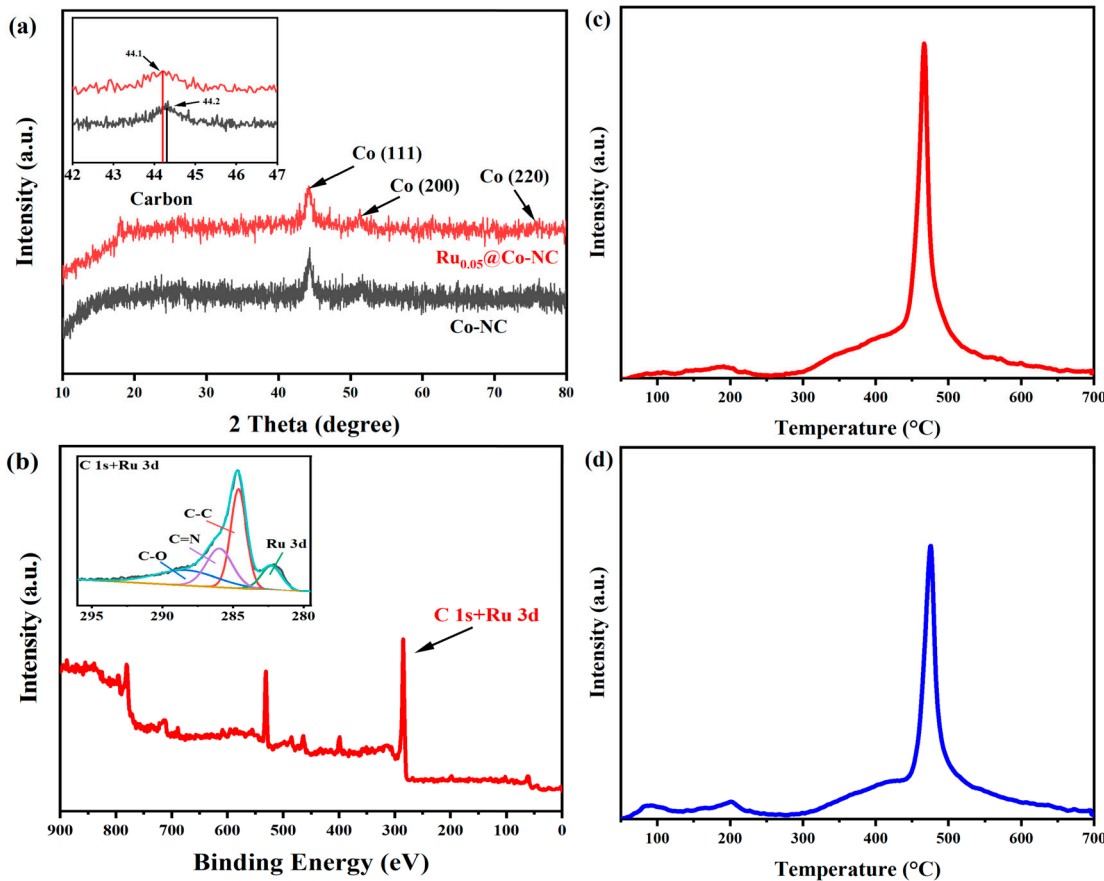

**Figure 2.** XRD diffractograms of (**a**) Co-NC and $Ru_{0.05}$@Co-NC; XPS spectra of (**b**) $Ru_{0.05}$@Co-NC; (**c**) $CO_2$ -TPD and (**d**) $NH_3$-TPD profiles of $Ru_{0.05}$@Co-NC.

Next, the morphological properties of Co-NC and $Ru_{0.05}$@Co-NC were evaluated. Upon examination by transmission electron microscopy (TEM), the morphologies of Co-NC (Figure 3a) and $Ru_{0.05}$@Co-NC (Figure 3b) were found to be almost identical, which revealed that the morphology of Co-NC was retained after the incorporation of Ru. It was also observed that the metal particles were embedded in nitrogen-doped carbon and surrounded by a graphitic carbon layer. The presence of nitrogen-doped graphitic carbon was believed to effectively inhibit cobalt particle aggregation during high-temperature pyrolysis while promoting metal species dispersion to ensure an evenly distributed position for Ru [47]. Moreover, energy dispersive X-ray (EDX) elemental mapping images unequivocally supported the uniform distribution of the Co, Ru, C, O, and N elements (Figure 3c–h). It was observed that the Co and Ru elements were uniformly distributed and enveloped by the nitrogen-doped carbon layer. Probably, the perpetual presence of Ru on the surface is due to the partial replacement of Co. It should be noted that the occurrence of the O element was also observed in related studies [28,33]. It is likely that the precursor material (ZIF-67) could absorb trace amounts of methanol (for the preparation and purification of ZIF-67) as well as moisture and oxygen (from the air). Following high-temperature treatment, it was surmised that the aforementioned oxygen sources reacted with other substances to produce O-containing species (such as Co–O, C=O, and N–O), which is consistent with the observations from the related catalysts in the literature [18,23].

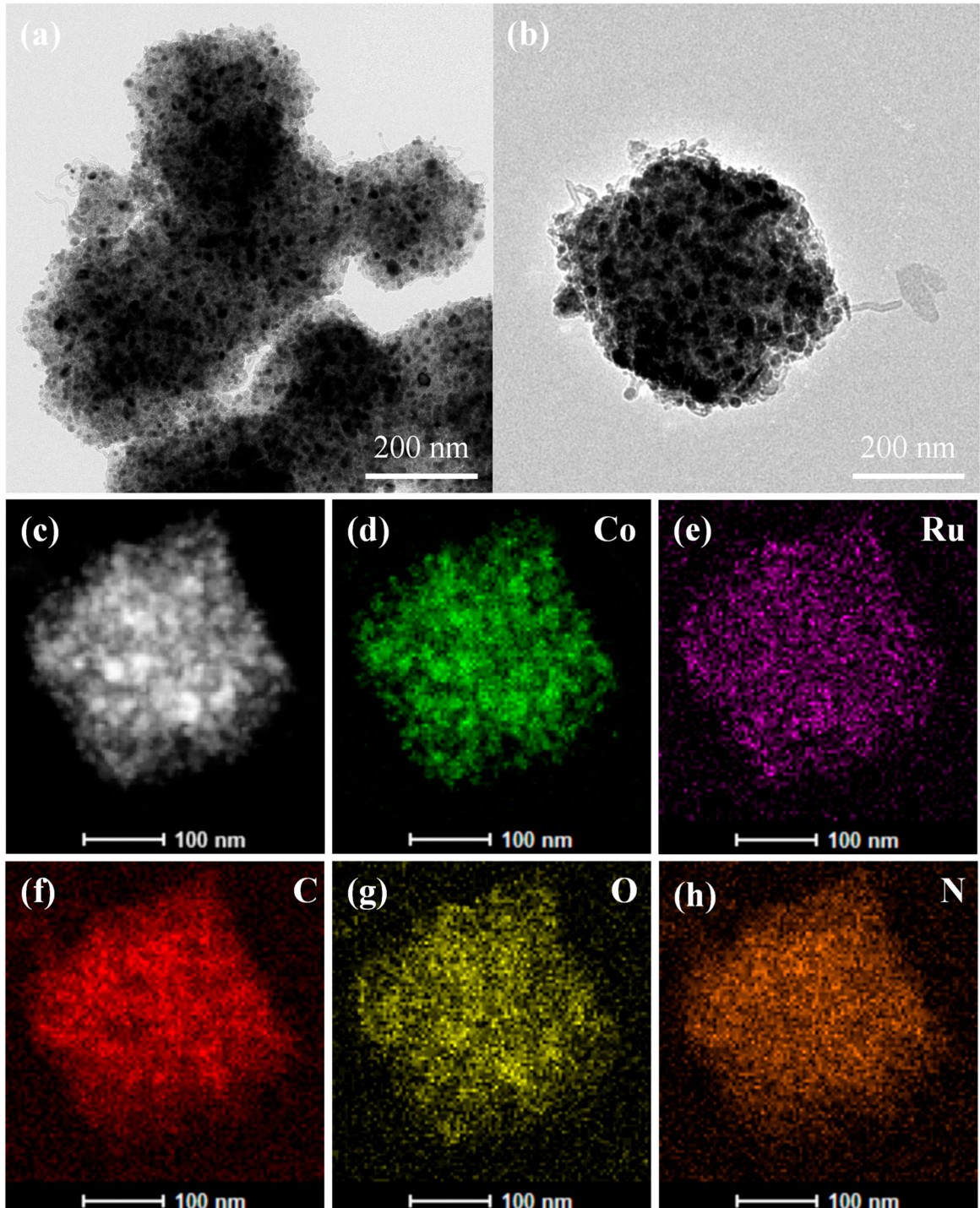

**Figure 3.** TEM images of (**a**) Co-NC, (**b**) Ru$_{0.05}$@Co-NC, and (**c–h**) the corresponding elemental mapping images of Ru$_{0.05}$@Co-NC.

On the other hand, the pore structure and properties of Ru$_{0.05}$@Co-NC were examined using the Brunauer Emmett Teller (BET) method to determine the specific surface area and pore size. The N$_2$ adsorption and desorption curve for Ru$_{0.05}$@Co-NC, as shown in Figure 4a, indicated that this material had strong adsorption capacity at low pressure (P/P$_0$ < 0.05), and a hysteresis loop was also observed at high pressure (P/P$_0$ = 0.8–0.99). The curve is of the type IV isotherm, indicating that the material is mainly microporous with some mesopores. The presence of these pore structures facilitates the operation of substrate molecules and their interaction with the active catalytic centers, thus promoting

this catalytic process [48]. Subsequently, the pore size distribution of the materials was explored and shown in Figure 4b. It was also observed that the curve of Co-NC was similar to that of $Ru_{0.05}$@Co-NC, which indicated that the pore structure of Co-NC was inherited despite the replacement of some Co species by Ru during the synthesis of $Ru_{0.05}$@Co-NC. The results showed that the micropore percentage of $Ru_{0.05}$@Co-NC was reduced compared with that of Co-NC, whereas the mesopore percentage of $Ru_{0.05}$@Co-NC was slightly higher than that of Co-NC. The possible reason is that the involvement of Ru occupied some of the micropores in the precursor material [49]. Furthermore, the specific surface area, pore volume, and pore diameter of the $Ru_{0.05}$@Co-NC and Co-NC material were determined (as outlined in Table S4), which illustrated that the introduction of Ru led to a decrease in the specific surface area and increases in pore volume and pore size. Probably, the incorporated Ru species could occupy the micropores of the precursor material (Co-NC). The decrease in the specific surface area is generally regarded as an indication of successful Ru introduction, while the occupation of the micropores also enhances the pore volume and pore size [49].

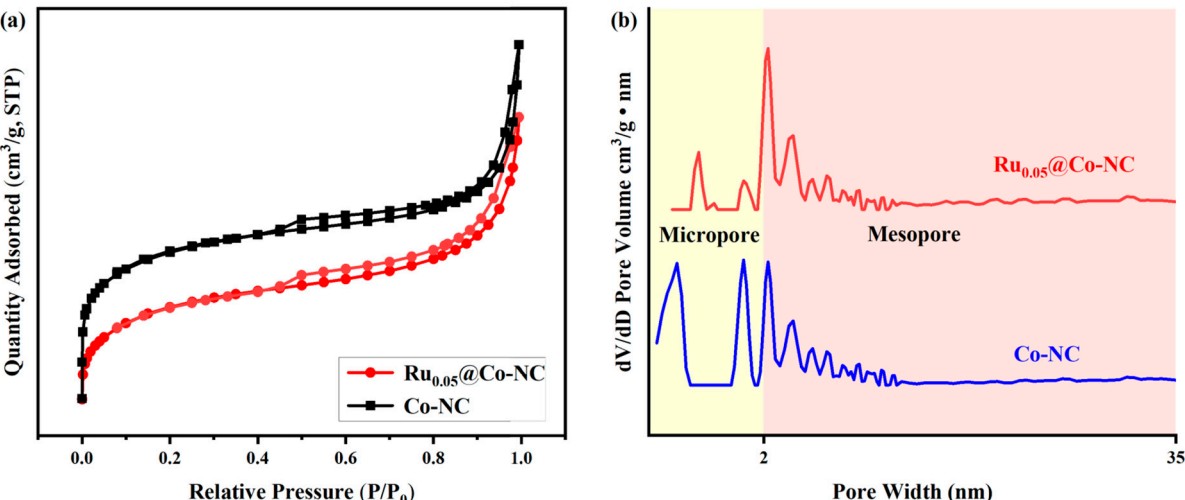

**Figure 4.** (**a**) Nitrogen adsorption-desorption isotherms at 77 K for Co-NC and $Ru_{0.05}$@Co-NC; (**b**) The pore-size distribution curves for Co-NC and $Ru_{0.05}$@Co-NC.

### 2.2. The Catalytic Activity and Recyclability of $Ru_x$@Co-NC

After the successful preparation of the required $Ru_x$@Co-NC materials, their catalytic performance was evaluated using the treatment of benzyl alcohol (**1a**) and KOH as a template reaction (Table 1). Initially, a stirred mixture of **1a** (2.00 mmol), KOH (2.40 mmol), $Ru_{0.05}$@Co-NC (5 mg), and toluene (0.5 mL) was heated at 120 °C under an argon atmosphere for 16 h, affording a 45% yield of **2a** (entry 1). It appeared that the concentration of the Ru precursor ($RuCl_3 \cdot 3H_2O$) used to prepare the target $Ru_x$@Co-NC materials marginally affected the catalytic performance, as evidenced by the fact that 0.05–0.20 M of $RuCl_3 \cdot 3H_2O$ provided comparable product yields ranging from 47 to 55% (entries 1–4). Thus, 0.05 M of $RuCl_3 \cdot 3H_2O$ was used for further investigations. The extension of the reaction time to 24 h afforded product **2a** in an enhanced yield of 65% (entry 5), and reducing the substrate amounts provided an even higher yield of 85% (entry 6). Next, different solvents were attempted (entries 1, 7–10), and the results indicated that toluene was identified as the optimal solvent. It was found that the solvent volume influenced the efficiency of this catalytic process, while either reducing or increasing the solvent volume led to inferior results (entry 6 vs. entries 11–12). Moreover, increasing the catalyst amount from 5 mg to 7.5 mg and 10 mg could not efficiently improve the product yield (entries 13–14), so 5 mg was still selected for further studies. To illustrate the pivotal role of Ru in this catalytic process, the template Co-NC and no catalyst were evaluated (entries 15–16). Co-NC produced only a moderate yield of **2a** and no catalyst led to trace amounts of **2a**, which proved the significant role of Ru in this transformation. Finally, the reaction in the absence of KOH

was carried out, providing only a trace amount of product **2a** (entry 17). Altogether, the optimized conditions were identified as those listed in entry 6.

**Table 1.** Optimization of reaction conditions [a].

$$Ph\diagdown OH + KOH \xrightarrow[\text{solvent, } y \text{ mL, } 120\,°C, \text{ Ar, } n \text{ h}]{\text{Catalyst (5 mg)}} \xrightarrow{\text{3 N HCl}} Ph\overset{O}{\underset{}{\diagdown}}OH$$

**1a**　　　　　　　　　　　　　　　　　　　　　　　　　　　　　　　　　　　**2a**

| Entry | Catalyst | Solvent | y | n | Conversion (%) [b] | Yield (%) [b] |
|---|---|---|---|---|---|---|
| 1 [a] | $Ru_{0.05}$@Co-NC | toluene | 0.5 | 16 | 51 | 50 |
| 2 [a] | $Ru_{0.1}$@Co-NC | toluene | 0.5 | 16 | 49 | 48 |
| 3 [a] | $Ru_{0.15}$@Co-NC | toluene | 0.5 | 16 | 53 | 49 |
| 4 [a] | $Ru_{0.2}$@Co-NC | toluene | 0.5 | 16 | 50 | 47 |
| 5 [a] | $Ru_{0.05}$@Co-NC | toluene | 0.5 | 24 | 68 | 65 |
| 6 [c] | $Ru_{0.05}$@Co-NC | toluene | 0.5 | 24 | 86 | 85(83 [d]) |
| 7 [a] | $Ru_{0.05}$@Co-NC | 1,4-dioxane | 0.5 | 24 | 73 | 68 |
| 8 [a] | $Ru_{0.05}$@Co-NC | *m*-xylene | 0.5 | 24 | 82 | 78 |
| 9 [a] | $Ru_{0.05}$@Co-NC | water | 0.5 | 24 | 15 | 9 |
| 10 [a] | $Ru_{0.05}$@Co-NC | - | 0.5 | 24 | 30 | 28 |
| 11 [c] | $Ru_{0.05}$@Co-NC | toluene | 0.25 | 24 | 74 | 72 |
| 12 [c] | $Ru_{0.05}$@Co-NC | toluene | 0.75 | 24 | 77 | 75 |
| 13 [c,e] | $Ru_{0.05}$@Co-NC | toluene | 0.5 | 24 | 85 | 84 |
| 14 [c,e] | $Ru_{0.05}$@Co-NC | toluene | 0.5 | 24 | 89 | 86 |
| 15 [c,f] | Co-NC | toluene | 0.5 | 24 | 51 | 48 |
| 16 [c] | - | toluene | 0.5 | 24 | 6 | 5 |
| 17 [g] | $Ru_{0.05}$@Co-NC | toluene | 0.5 | 24 | 4 | <1 |

[a] **1a** (2.0 mmol), KOH (2.4 mmol) and catalyst (5 mg) were heated at reflux under an argon atmosphere; [b] nuclear magnetic resonance (NMR) yields with 1,3,5-trimethoxybenzene as an internal standard and $CDCl_3$ as a solvent; [c] **1a** (1.0 mmol), KOH (1.2 mmol); [d] isolated yields; [e] catalyst (7.5 mg); [f] catalyst (10.0 mg); [g] **1a** (1.0 mmol), no KOH.

The catalytic potential of the $Ru_{0.05}$@Co-NC-mediated protocol was investigated by determining the scope of suitable substrates under the optimized conditions (Scheme 1). Firstly, benzyl alcohols with a Me group at different positions were chosen for analysis to examine the effect of steric hindrance on the reactions. It was observed that no obvious steric effects were detected for the substrates (**1b–1d**), as comparable yields of 73%, 74%, and 65% were obtained for the corresponding carboxylic acid products **2b**, **2c**, and **2d**. Secondly, the influence of the electronic characteristics on the activity of $Ru_{0.05}$@Co-NC was investigated. It was found that the electron-rich alcohols **1d–1e** produced significantly higher reactivity compared to the electron-poor counterparts **1f–1g**, as evidenced by the isolated yields of 71–81% for acids **2d–2e** and only 20–25% for products **2f–2g**. Apparently, aromatic compounds containing heteroatoms (**1h–1i**) were studied as viable substrates, leading to acid products (**2h–2i**) in 60–70% yields. Similarly, alcohols **1j–1k** that contain a diphenyl or naphthyl group were found to serve as suitable coupling partners and can be smoothly converted into the targeted acid products (**2j–2k**) with 65–75% yields. Apart from aromatic alcohols, aliphatic counterparts (**1l–1n**) were also evaluated. It was obvious that the catalytic system was susceptible to the steric effects incorporated into the aliphatic alcohols. This conclusion was supported by the fact that sterically unhindered alcohols **1l-1m** delivered products **2l–2m** in 65–73% yields, whereas the congested substrate (**1n**) yielded acid **2n** in only 51%. The conversion of amino alcohols into their respective amino acids was also performed, producing the required acid (**2o**) in a 71% yield. Altogether, this approach presents promising potential in the synthesis of various carboxylic acids.

**Scheme 1.** Exploration of the substrate scope.

Subsequently, the recovery and recycling of $Ru_{0.05}$@Co-NC were carried out by utilizing the conversion of **1a** into **2a** for the investigations. To better demonstrate the recyclability of $Ru_{0.05}$@Co-NC, the selected conditions (with more substrates and a shorter reaction time compared to the standard conditions) were utilized to ensure a product yield of no more than 50% for the first catalytic cycle. As shown in Figure 5, it was gratifying that the catalyst displayed continued recyclability for eight cycles, with each cycle producing **2a** in a reliable moderate yield. Moreover, it was observed that 1 mg of $Ru_{0.05}$@Co-NC effectively produced 183 mg of **2a** after eight cycles, almost twice as much as the value from Co-NC (90.9 mg) after nine cycles (Figure S1) [23]. These findings highlight the significant potential of the $Ru_{0.05}$@Co-NC catalyst for further studies and applications.

*2.3. Stability of $Ru_{0.05}$@Co-NC*

The best-performing catalyst ($Ru_{0.05}$@Co-NC) was examined for its structural and morphological properties before and after the reaction, which are important indicators of a heterogeneous catalyst. XRD diffractograms exhibited no obvious changes between the fresh and recycled $Ru_{0.05}$@Co-NC (Figure 6a), which indicated that the crystal structure of the catalyst did not vary after the reaction. When comparing the surface elemental compositions before and after the reaction, we observed a decrease in the Ru content but no obvious changes in the Co content (Table S2). Despite these changes in the elemental composition, the catalytic effect remains almost unchanged after the reaction. Probably, only a small amount of Ru species could stably exist and play crucial roles in catalyzing this transformation, as discussed in detail in the following acid etching experiments. In the ICP results, only a minimal decrease was detected in the Ru and Co contents before and after the reaction (Table S3), which proved that $Ru_{0.05}$@Co-NC could stably maintain its properties after the reaction. Moreover, the morphologies of fresh and recycled catalysts were explored and compared (Figure 6b–e). Scanning electron microscopy (SEM) images revealed that the fresh $Ru_{0.05}$@Co-NC had a rhombic dodecahedral structure with a rough surface

(Figure 6b,c). Regarding the recycled catalyst, the surface morphology still maintained the typical rhombic dodecahedral structure, even though slightly rougher surfaces were detected (Figure 6d,e). TEM images also showed that the morphology of $Ru_{0.05}$@Co-NC did not obviously change after the reaction (Figure S2). Additionally, EDX mapping demonstrated that the individual elements remained uniformly distributed after the reaction (Figure S3). All the above results confirmed the structural and morphological stability of $Ru_{0.05}$@Co-NC, even under standard reaction conditions. Therefore, $Ru_{0.05}$@Co-NC can be considered a stable heterogeneous catalyst in this transformation.

　　　XPS characterization was performed to have a clear understanding of the chemical states of the main elements in both fresh and recycled $Ru_{0.05}$@Co-NC. It was clearly observed that the material mainly comprised Co, Ru, O, N, and C elements (Figure S4). The high-resolution XPS spectra of $Ru_{0.05}$@Co-NC before and after the reaction were determined and compared (Figure 7). For the C 1s spectra of both fresh and recycled catalysts, C–O, C=N, and C–C were identified as the main carbon species (Figure 7a,b). The typical $Ru^0$ 3d peak was also found in the recycled catalyst, indicating that the chemical state of Ru was still $Ru^0$ after the reaction. For the Co 2p spectra, three typical peaks located at 782.8 eV, 780.1 eV, and 778.4 eV were assigned to the Co $2p_{3/2}$ peaks of the Co–N, Co–O, and $Co^0$ species, respectively. Meanwhile, the corresponding Co $2p_{1/2}$ signals were positioned at 797.9 eV, 795.2 eV, and 793.4 eV, respectively. Along with these peaks, two satellite peaks were also detected [50–52]. Subsequently, the fresh and used catalysts were compared, which indicated a similar composition of the Co species, despite differences in the ratio (Figure 7c vs. Figure 7d). Furthermore, the main N species showed a similar trend both before and after the reaction (Figure 7e,f). Based on the above results, we concluded that the chemical states of the main elements in $Ru_{0.05}$@Co-NC did not vary after the reaction, which further revealed the stability of this catalyst under the standard conditions.

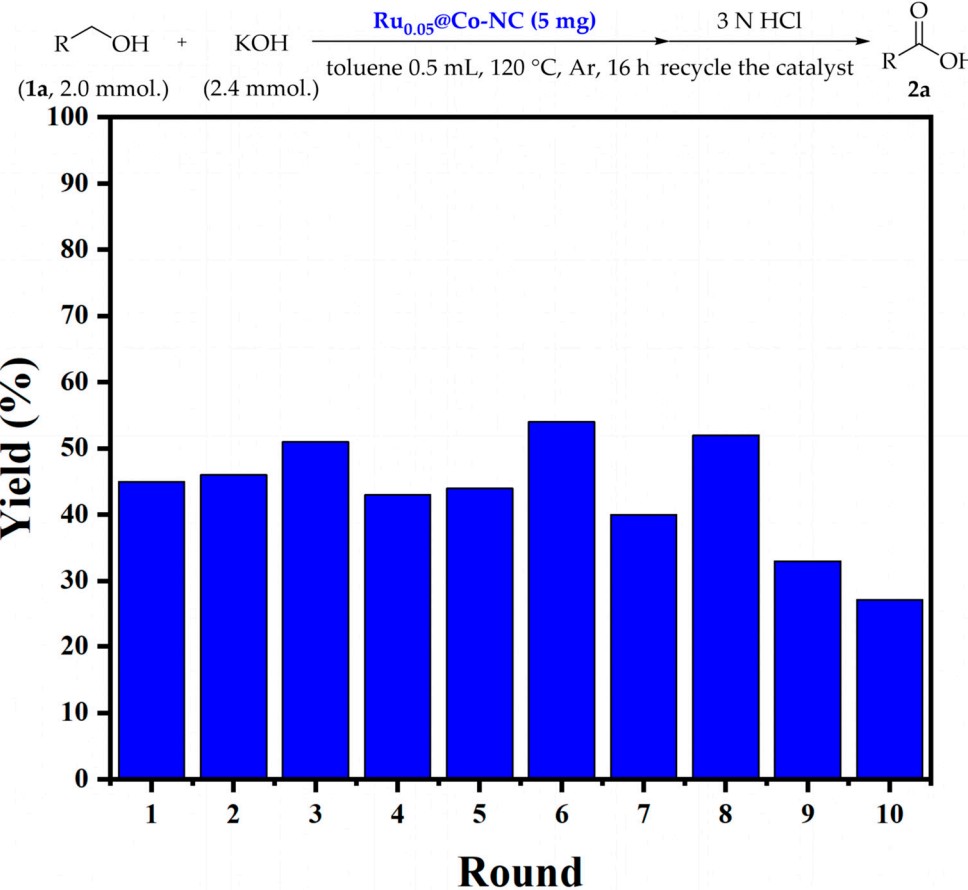

**Figure 5.** Recyclability test of $Ru_{0.05}$@Co-NC. Note: 5 mg of the catalyst was added only in the first run, while no catalyst was added in the other runs.

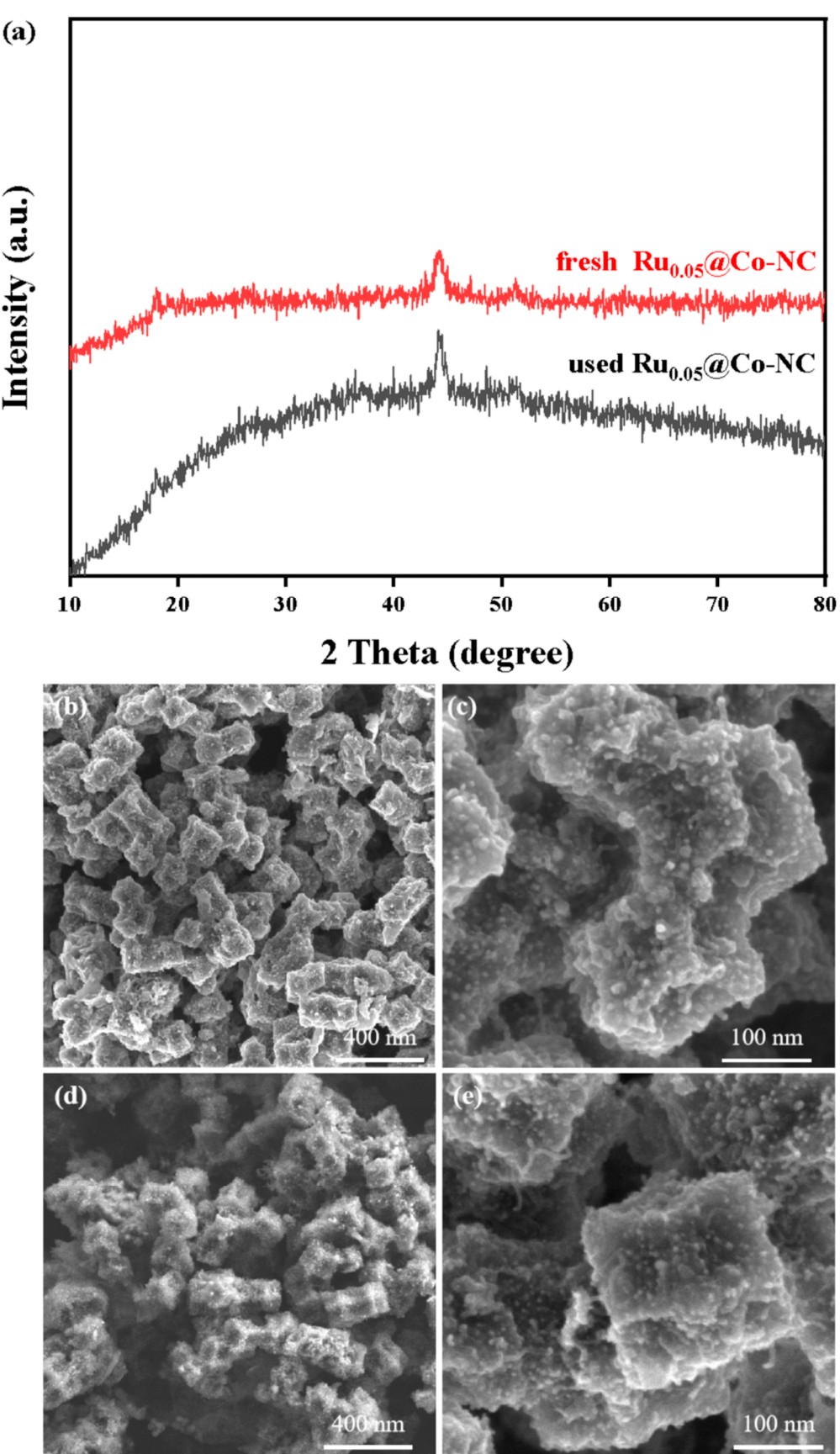

**Figure 6.** (**a**) XRD diffractograms of fresh $Ru_{0.05}@Co-NC$ and used $Ru_{0.05}@Co-NC$; SEM images of (**b**,**c**) fresh $Ru_{0.05}@Co-NC$,(**d**,**e**) used $Ru_{0.05}@Co-NC$.

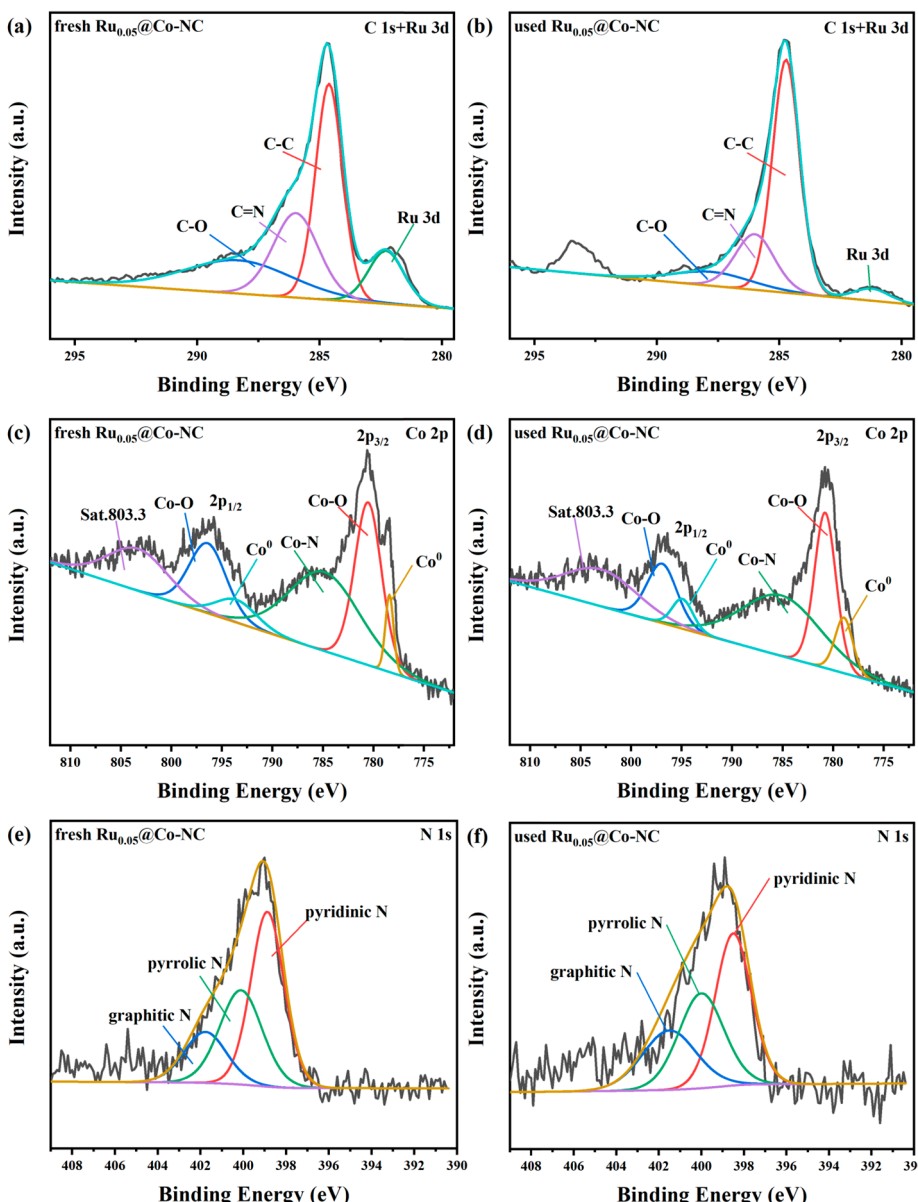

**Figure 7.** XPS spectra in C1s+Ru 3d of (**a**) fresh Ru$_{0.05}$@Co-NC and (**b**) used Ru$_{0.05}$@Co-NC; (**c**) Co 2p of fresh Ru$_{0.05}$@Co-NC and (**d**) used Ru$_{0.05}$@Co-NC; N 1s of (**e**) fresh Ru$_{0.05}$@Co-NC and (**f**) used Ru$_{0.05}$@Co-NC.

The stability of the incorporated Ru$^0$ species in Ru$_{0.05}$@Co-NC was evaluated by etching the catalyst with concentrated acids, resulting in Ru$_{0.05}$@Co-NC-HCl and Ru$_{0.05}$@Co-NC-HNO$_3$. For comparative purposes, Co-NC-HCl and Co-NC-HNO$_3$ were also obtained via the acid etching of the Co-NC precursor material. With the above materials in hand, their catalytic performance was evaluated and compared (as listed in Table 2). The catalytic activity of Ru$_{0.05}$@Co-NC-HCl remained largely unchanged after HCl etching (entry 2), while Ru$_{0.05}$@Co-NC-HNO$_3$ produced a considerably lower yield of **2a** (entry 3). Similar results were obtained for Co-NC etched with the corresponding acids (entries 4–6). Importantly, the inclusion of Ru always triggered higher catalytic activity compared to the corresponding Ru-free precursors (entry 1 vs. entry 4, entry 2 vs. entry 5, entry 3 vs. entry 6). This finding suggested that Ru incorporation played a vital role in promoting the catalytic reaction.

**Table 2.** Catalytic performance of the etched catalysts [a].

R—OH + KOH $\xrightarrow[\text{toluene, 0.5 mL, 120 °C, Ar, 24 h}]{\text{Catalyst (5 mg)}}$ $\xrightarrow{\text{3 N HCl}}$ R—C(=O)OH

(**1a**, 1.00 equiv.)    (1.20 equiv.)                  **2a**

| Entry [a] | Catalyst | Conversion (%) [b] | Yield (%) [b] |
|:---:|:---:|:---:|:---:|
| 1 | $Ru_{0.05}$@Co-NC | 86 | 85 |
| 2 | $Ru_{0.05}$@Co-NC-HCl | 84 | 82 |
| 3 | $Ru_{0.05}$@Co-NC-HNO$_3$ | 54 | 51 |
| 4 | Co-NC | 51 | 48 |
| 5 | Co-NC-HCl | 54 | 50 |
| 6 | Co-NC-HNO$_3$ | 22 | 21 |

[a] **1a** (2.0 mmol), KOH (2.4 mmol) and catalyst (5 mg) were heated at reflux under an argon atmosphere; [b] NMR yields with 1,3,5-trimethoxybenzene as an internal standard and CDCl$_3$ as a solvent.

To explain the different effects of conc. HCl and HNO$_3$ on the catalytic performance of the materials, additional experiments were conducted. SEM images revealed that no obvious changes were observed after acid etching (Figure S5). The XPS spectrum of $Ru_{0.05}$@Co-NC-HCl was almost identical to that before etching (Figures S4 and S6), but ICP measurements revealed a decrease of nearly 3% in both Co and Ru contents (Table S3). The XPS patterns confirmed the presence of the same C and Co species in the material after HCl etching (Figure 8a,c). Although the Ru 3d peak almost disappeared, the peak of Ru 3p could still be observed (Figure 8e), indicating that a small amount of Ru$^0$ remained in the material, which is consistent with the ICP results. It was possible that conc. HCl only removed unstable metal nanoparticles without destroying the overall catalyst structure, resulting in a comparable catalytic performance between $Ru_{0.05}$@Co-NC and $Ru_{0.05}$@Co-NC-HCl. In contrast, the etching process using conc. HNO$_3$ significantly affected the material structure (Figure S5). Specifically, an additional π-π peak was detected in $Ru_{0.05}$@Co-NC-HNO$_3$ (Figure 8b), which may be generated due to the electron transfer caused by structural damage to the carbon material from concentrated nitric acid [53]. It was also found that the Co peaks in the XPS pattern of $Ru_{0.05}$@Co-NC-HNO$_3$ almost disappeared (Figure 8d), indicating that most of the Co species were removed. The disappearance of the Ru 3d peak and the presence of Ru$^0$ at the Ru 3p position (Figure 8f) also confirmed the structural damage caused by HNO$_3$ etching. The substantial decrease in the yield of **2a** after HNO$_3$ etching was likely due to the aforementioned structural damage. The above explanations were the likely reasons for the considerably lower yield of **2a** after HNO$_3$ etching. Interestingly, despite being treated with conc. HNO$_3$, the Ru content in $Ru_{0.05}$@Co-NC-HNO$_3$ was similar to that of $Ru_{0.05}$@Co-NC-HCl, suggesting that some Ru species could stably exist in strong acid environments.

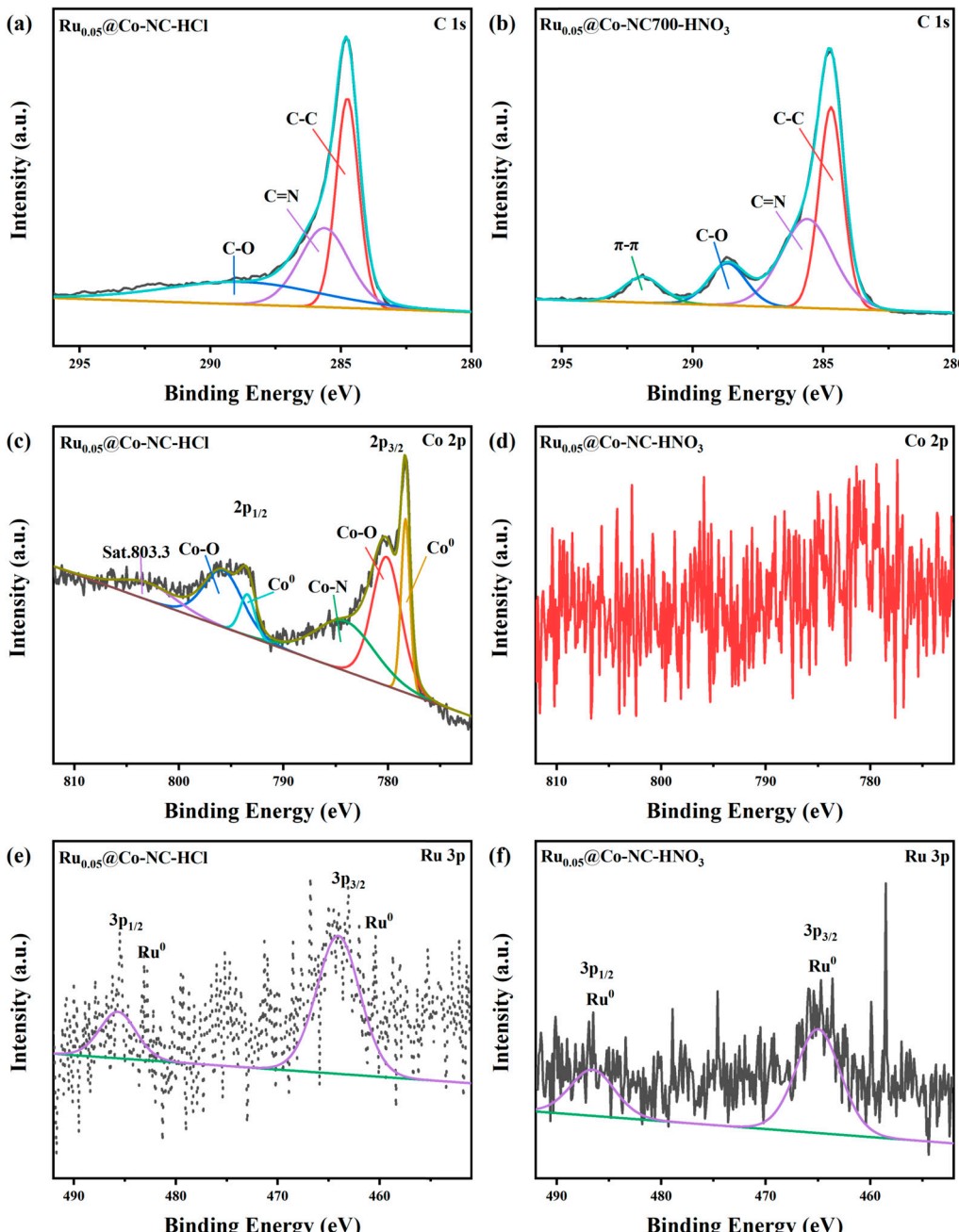

**Figure 8.** XPS spectra in C 1s + Ru 3d of (**a**) Ru$_{0.05}$@Co-NC-HCl and (**b**) Ru$_{0.05}$@Co-NC-HNO$_3$; Co 2p of (**c**) Ru$_{0.05}$@Co-NC-HCl and (**d**) Ru$_{0.05}$@Co-NC-HNO$_3$; Ru 3p of (**e**) Ru$_{0.05}$@Co-NC-HCl and (**f**) Ru$_{0.05}$@Co-NC-HNO$_3$.

## 3. Experimental Section

### 3.1. Materials Preparation

#### 3.1.1. Preparation of ZIF-67

The synthetic procedure of ZIF-67 was identical with that in a previously reported work [23], as described in the following texts: Co(NO$_3$)$_2$·6H$_2$O (0.58 g, 2.0 mmol) was dissolved in methanol (25 mL) to form a solution. Subsequently, a solution of 2-methylimidazole (2-MIM, 1.32 g, 16.0 mmol) in methanol (25 mL) was sonicated for 5 min, and then added to the Co(NO$_3$)$_2$·6H$_2$O solution while stirring at 600 rpm. The resulting mixture was swirled at room temperature for 24 h to generate a suspension, which was centrifuged at 8000 rpm (5 min × 3). After that, the collected precipitate was washed with methanol (20 mL × 3)

before being dried at a temperature of 80 °C under vacuum for 24 h. The final product obtained was a purple ZIF-67 powder.

### 3.1.2. Preparation of Co-NC, Co-NC-HCl, and Co-NC-HNO$_3$

The synthesized ZIF-67 was distributed evenly in a ceramic boat, and it was then pyrolyzed at 700 °C with a flow rate of 50 mL/min under an argon atmosphere containing 5% hydrogen gas (H$_2$/Ar). The Co-NC was obtained as a black powder.

Co-NC-HCl was obtained by the etching of Co-NC with concentrated hydrochloric acid (conc. HCl) at room temperature for 12 h, while Co-NC-HNO$_3$ was afforded by the treatment of Co-NC with 9 M nitric acid (9 M HNO$_3$) at 120 °C for 4 h.

### 3.1.3. Synthesis of Ru@Co-NC, Ru@Co-NC-HCl and Ru@Co-NC-HNO$_3$

Co-NC (50 mg) was dispersed in 20 mL of deionized water and stirred at 1000 rpm for 30 min. Next, 1 mL of a pre-prepared RuCl$_3$·3H$_2$O (0.05 mol/L, 0.1 mol/L, 0.15 mol/L, or 0.2 mol/L) solution was added, and the mixture was stirred at the same speed at room temperature for 24 h. The mixture was filtered to extract the black powder, Ru$_x$@Co-NC (x = 0.05, 0.1, 0.15, 0.2), which was dried overnight in a vacuum oven at 80 °C. The Ru loadings of these materials were expected to be 2–8 wt%. Moreover, the optimized Ru$_x$@Co-NC was etched in the same manner with conc. HCl or 9 M HNO$_3$ to produce Ru$_x$@Co-NC-HCl or Ru$_x$@Co-NC-HNO$_3$.

### 3.2. Catalyst Characterization

To determine the crystal structure of the materials, powder XRD was conducted using a Bruker D8 Advance X-ray diffractometer (with a copper radiation source, Karlsruhe, Germany). The surface area, gas adsorption, and porosity properties were measured by BET on the Micromeritics ASAP 2020 instrument (Micromeritics, Norcross, GA, USA), with the use of N$_2$ as probe gas. In addition, the samples were activated at 70 °C under vacuum for 3 h before the measurements. The materials' morphologies were investigated by SEM (TESCAN MIRA3/Tescan Mira4, Brno, Czech Republic) and TEM (FEI Tecnai G2F30, Hillsboro, OR, USA) using a JSM-6700F field emission scanning electron microscope (FESEM) instrument and JEOL JEM-2100F TEM apparatus (Tokyo, Japan), respectively. To determine the mapping of the elements, an EDAX Genesis instrument was used for an EDX. The Thermo Fischer ESCALAB 250Xi spectrometer (Waltham, MA, USA) was used to conduct XPS investigations. The elemental composition of the catalyst was determined using ICP-OES (Prodigy 7, Hudson, NH, USA) studies. Each sample was placed in a high-pressure reactor and pre-treated with a mixture of aqua regia, hydrofluoric acid, and hydrogen peroxide at 300 °C until the formation of a uniform solution. An AutoChem II 2920 V5.02 apparatus (Micromeritics, Norcross, GA, USA)) was used to perform TPD tests, with the use of NH$_3$/CO$_2$ as the probe gas to evaluate the acid/basic properties of the catalyst. NMR analyses were used to calculate the product yields, and the spectra were recorded with a Bruker Avance 500 spectrometer ($^1$H NMR at 500 MHz and $^{13}$C NMR at 126 MHz) using CDCl$_3$, DMSO-$d_6$, or D$_2$O as a deuterated solvent. Multiplicity was designated by the following abbreviations: s = singlet, d = doublet, t = triplet, and m = multiplet.

### 3.3. Catalytic Reactions

In a 25 mL Schlenk tube, Ru$_x$@Co-NC (5 mg), KOH (67.3 mg, 1.2 mmol), benzyl alcohol (104 μL, 1.0 mmol), and toluene (0.5 mL) were added. The tube was placed under an argon atmosphere and then heated at 120 °C for 24 h. Once the mixture was cooled to ambient temperature, the catalyst was separated with a magnet, and then washed with water (2 mL × 1) and methanol (5 mL × 3). The methanol solution was concentrated and combined with the aqueous solution, and the resulting mixture was acidified by a 3 N HCl solution to reach a pH value of around 2. Afterwards, the acidified mixture was extracted with ethyl acetate (5 mL × 3). The combined organic layers were dried with anhydrous

$Na_2SO_4$ and concentrated under reduced pressure. Finally, the methods for determining the conversion, NMR yields, and isolated yields of all the acid products were similar with those reported in our previous work [27].

### 3.4. Catalyst Recycling

For catalyst recycling, **1a** (2 mmol), KOH (2.4 mmol), and $Ru_{0.05}$@Co-NC (5 mg) were heated at refluxing toluene (0.5 mL) under an argon atmosphere for 16 h. The selection of these conditions was to ensure that the product yield was no more than 50%. After the reaction, an external magnet was used to separate the catalyst from the mixture, and the recovered catalyst was then washed with water (10 mL × 1) and methanol (10 mL × 3). The washed catalyst was dried under vacuum overnight. The following runs adhere to the same process. In particular, just 5 mg of the catalyst was applied to the first cycle, while no additional catalyst was added to the other catalytic cycles.

### 4. Conclusions

Herein, a few Ru-based materials were designed and prepared by introducing the Ru species into a ZIF-67-derived Co-NC material using an efficient and facile approach. Among the as-prepared materials, $Ru_{0.05}$@Co-NC demonstrated the highest activity for the alcohol dehydrogenation to carboxylic acids. Furthermore, this catalyst could be recycled for eight consecutive runs, while maintaining consistent yields of approximately 50%. It is worth noting that the catalyst could be conveniently recovered using an external magnetic field, and directly reused without any special treatment or activation. The fresh and reused $Ru_{0.05}$@Co-NC catalysts were characterized by using several characterization techniques, which confirmed the high stability of the catalyst when subjected to the standard reaction conditions. Furthermore, acid etching experiments verified the outstanding stability of some Ru species within $Ru_{0.05}$@Co-NC, which could be crucial for the high activity and recyclability of the catalyst.

**Supplementary Materials:** The following supporting information can be downloaded at: https://www.mdpi.com/article/10.3390/catal13081225/s1, Table S1. Comparison of the present work with literature reports for the catalytic dehydrogenative of alcohols to carboxylates; Table S2. The elemental atomic surface percentage by XPS of fresh $Ru_{0.05}$@Co-NC, used $Ru_{0.05}$@Co-NC, $Ru_{0.05}$@Co-NC-HCl, $Ru_{0.05}$@Co-NC-HNO$_3$; Table S3. The elemental content determined by ICP of fresh $Ru_{0.05}$@Co-NC, used $Ru_{0.05}$@Co-NC, $Ru_{0.05}$@Co-NC-HCl, $Ru_{0.05}$@Co-NC-HNO$_3$, Co-NC, Co-NC-HCl, and Co-NC-HNO$_3$; Table S4. The BET surface areas, Langmuir surface areas, pore volume, and pore sizes of Co-NC and $Ru_{0.05}$@Co-NC; Figure S1. The amount of benzoic acid produced by Co-NC/$Ru_{0.05}$@Co-NC per round and cumulative yield per mg of catalyst; Figure S2. TEM images of fresh $Ru_{0.05}$@Co-NC and used $Ru_{0.05}$@Co-NC; Figure S3. The TEM elemental mapping images of used $Ru_{0.05}$@Co-NC; Figure S4. XPS spectra of $Ru_{0.05}$@Co-NC, $Ru_{0.05}$@Co-NC-HCl, and $Ru_{0.05}$@Co-NC-HNO$_3$; Figure S5. SEM images of $Ru_{0.05}$@Co-NC and $Ru_{0.05}$@Co-NC-HCl; (c) and $Ru0.05$@Co-NC-HNO$_3$; Figure S6. XPS spectra in N 1s of $Ru_{0.05}$@Co-NC-HCl and $Ru_{0.05}$@Co-NC-HNO$_3$; The characterization data for compounds **2a–2o**; Original $^1$H and $^{13}$C NMR spectra for compounds **2a–2o**; References.

**Author Contributions:** Conceptualization, Z.C. and J.H.; data curation, Z.C., J.H. and S.Z.; writing—original draft preparation, Z.C.; writing—review and editing, F.V., C.C. and Y.Y.; visualization, Z.C. and S.Z.; project supervision, C.C. and Y.Y; funding acquisition, C.C. and Y.Y. The manuscript has been read and revised by all authors before submission. All authors have read and agreed to the published version of the manuscript.

**Funding:** This research was supported by the Natural Science Foundation of Hubei Province (No. 2022CFB388), the Natural Science Foundation of Hainan Province of China (Grant No. 623MS068), the National Natural Science Foundation of China (No. 22102127).

**Data Availability Statement:** Not applicable.

**Acknowledgments:** We would also like to thank eceshi (www.eceshi.com) for the TPD tests.

**Conflicts of Interest:** The authors declare no conflict of interest.

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
