# Peer review of "Zeolitic Imidazolium Frameworks-Derived Ru-Based Composite Materials Enable the Catalytic Dehydrogenation of Alcohols to Carboxylic Acids"

_catalysts, doi:10.3390/catal13081225_

Round 1

Reviewer 1 Report

The research work presented in this manuscript addresses the challenge to develop active and stable heterogeneous catalysts for dehydrogenative conversion of alcohols to carboxylic acids. According to the literature, the co-presence of Lewis acid sites and basic centres is beneficial to the reaction. For this reason, authors proposed the combination of metallic Ru(0) Lewis acid sites and a Co-modified porous N-doped carbon (Co-NC), which should expose surface basic sites. The Co-NC support was prepared by pyrolysis of zeolitic imidazolium framework ZIF-67. The effect of Ru content was investigated. The resulting catalysts were characterized in their main morphological and structural properties by several techniques. The topic might be relevant in the field of heterogeneous catalyst design for fine chemical production. In this field, extensive research is active, as also reported by authors, who cited appropriate references. Recently, Co-NC catalysts have been reported to be effective in the base-free oxidation of alcohols to esters at room temperature and atmospheric conditions. The novelty of the present contribution lies in the use of Ru. However, despite the modest novelty, the choice of using Ru may be controversial, considering the high cost of this metal. Moreover, the adopted experimental conditions (Ar atmosphere and addition of KOH) pose severe problems in terms of practical applications. In general, there are several critical points that affect the quality of the proposed research and make the paper not suitable for publication in Catalysts (see comments below).

1.     Abstract. In the abstract Co-NC is given without full name at its first appearance. It could be preferable to cite some relevant quotes in the abstract (e.g. concerning recycles, authors should quantify the cited “little activity decay”).

2.     It is suggested to improve keywords, avoiding repeating terms already reported in the title.

3.     The description of Ru@Co-NC catalysts does not contain all the useful details (which volume of metal solution was used? What about the expected Ru loading?)

4.     Details about sample preparation and pre-treatment conditions for surface area measurements and ICP analyses (e.g. digestion procedure) must be reported. Please, specify in the experimental that N2 was used as probe gas for surface area measurements.

5.     The Reviewer cannot trace the reaction temperature for catalytic tests in the manuscript.

6.     Figure 1 does not contain enough details to be readable and clearly comprehensible. Such a figure is likely more appropriate as a graphical abstract.

7.     P.5. Please, authors are invited to use the correct terminology. E.g., the experimental outcome from XRD analysis must be indicated as diffractogram (instead of spectrum as authors erroneously wrote).

8.     Among the causes for the lack of detectable Ru reflections in XRPD patterns, authors should mention also the very low Ru concentration (ca. 3.5 wt.%).

9.     “More importantly, a typical Ru0 3d peak at 282.1 eV was also found in the C 1s spectra”, please rephrase this ambiguous sentence.

10.  Concerning the surface composition of prepared catalysts, authors showed the survey spectra, but they did not report the surface composition in terms of %at. concentrations for all the detected elements. This is an important information that should help in interpreting the catalytic results. The same comment applies also for the characterization of used catalysts.

11.  It is suggested that the proportion of elements in TEM-EDX is added.

12.  Authors justify the presence of O in EDS maps by invoking the oxidation of Ru, Co, C and N by oxygen sources (water, O2 and methanol) during the thermal treatment. However, the thermal treatment was performed at 700°C under an argon atmosphere containing 5% hydrogen gas. Are these reducing conditions compatible with the hypothesized oxidative processes invoked by the authors?

13.  P-7, The explanation authors provided for the hysteresis loop appearance in the ads./des. Isotherm (i.e. “condensation of liquid nitrogen in the pores during desorption”) is questionable.

14.  According to data in Table S.2, the introduction of Ru seems to cause a remarkable decrease in SSA (from 200 to 145 m2 g-1), pore volume (from 0.16 to 0.08 cm3 g-1) and pore size (from 5 to 1 nm) of Co-CN. Nevertheless authors state that “the specific surface area, pore volume, and pore diameter of Ru0.05@Co-NC and Co-NC material were determined (as outlined in Table S2), which illustrated that these values in the two materials did not differ too much.“. Authors are invited to reconsider their statement.

15.  The presentation of catalytic results is not clear, too confusing and poor. It should be re-written. Results should be reported in terms of alcohol conversion and selectivity (or yield) to all the detected products. Moreover, comparison with literature data is a must.

16.  It is surprising that authors did not investigate the surface acid and base features of the studied catalysts. Considering the key role of Lewis acid sites and basic sites in the studied reaction, authors should deeply investigate the acidic/basic character of the catalysts. In addition, catalytic tests in base free conditions should be also presented.

Reviewer 2 Report

The manuscript describes an extended work connected to catalytic dehydrogenation of alcohols to carboxylic acids on Ru promoted ZIF-67 derived Co-NC catalysts. A large amount of work has been performed and evaluated, thus the manuscript certainly deserves publication.

However, there are a few points which may need some further clarification:
= The manuscript claims that presence of both Lewis acid and basic sites are essential for promoting the desired reactions. Were there the strengths of these sites ever quantified (eg. with pyridine or CO2 adsorption)?

= What is the reasonable explanation for that ca. of 1/3 the Co content of the Co-NC is lost during a mild Ru impregnation (comparison of entries 1 and 5 in table S1)?

= What is a reasonable explanation that the yields for the Ru0.05@Co-NC and Ru0.05@Co-NC-HCl  catalysts are almost the same (Table 2), and simultaneously the Ru content is less than one tenth in the latter catalyst (entries 1 and 3 in table S1)? (In particular that distinguished role is attributed to Ru in the process.)

= Similarly, the activity of Co@NC without any Ru is still ca. one fourth that of the Ru0.05@Co-NC catalysts’ (comparison of entries 6 and 15 in table 1). Is there any hint for the structure or functioning of the catalytic centres? (Maybe some additional synergism of Co and Ru can also be supposed.)

Reviewer 3 Report

Reviewing Report for the manuscript ID: catalysts-2504649

Here is my reviewing report for the manuscript under title “Zeolitic imidazolium frameworks-derived Ru-based composite materials enable the catalytic dehydrogenation of alcohols to carboxylic acids” for catalysts with ID number: catalysts-2504649.  Kindly find it.

In this manuscript, the authors reported preparation a series of Ru-containing heterogeneous catalysts with zeolitic imidazolium frameworks (ZIFs)-derived materials for the catalytic dehydrogenation of alcohols to carboxylic acids. The results showed that Ru0.05@Co-NC displayed the best catalytic performance in this transformation. Furthermore, the catalyst showed remarkable recyclability as evidenced by recycling eight times without high decade in conversion efficiency.

Overall, the study is interesting, and I recommend accepting the manuscript for publication in catalysts in the present form.

The authors need to revise their manuscript according to the following suggestions.

1-    The language of the manuscript is clear and well-presented.

2-    In the manuscript, Figure and Fig. use full name sometimes and in others use the abbreviations. Please carefully revise the whole manuscript.

3-    The number of citations in the manuscript is too many, the authors recommend minimizing the number to about 50.

All regards,

Round 2

Reviewer 1 Report

I carefully read the revised version of the manuscrip. Unfortunately, there are still some important points, which require further clarification.

1) It is not sufficient to report a comparative Table with literature data. Authors are invited to critically discuss the performance of their catalysts in comparison to other catalysts.

2) The characterization of acidic and basic features should be put in relation with catalytic results.

3) Surface composition from XPS should be commented and used to interpret the catalytic performance and stability (e.g. decrease in Ru surface content in the catalyst after use).

4) The catalytic tests in different solvents have been performed at different temperatures. This makes meaningless any comparison. 

5) In general authors are invited to discuss in a more critical way their results, trying to provide possible explaination for their observations (e.g. why the introduction of Ru leads to a decrease in specific surface area and increases in pore volume and pore size???)
